# The Effect of Direct-Fed Microbials on In-Vitro Rumen Fermentation of Grass or Maize Silage

**Rajan Dhakal** [1],*, **Giuseppe Copani** [2], **Bruno Ieda Cappellozza** [2], **Nina Milora** [2] and **Hanne Helene Hansen** [1],*

1   Department of Veterinary and Animal Sciences, University of Copenhagen, Grønnegårdsvej 3, 1870 Frederiksberg C, Denmark
2   Chr. Hansen Animal and Plant Health & Nutrition, Bøge Alle 10-12, 2970 Hørsholm, Denmark
*   Correspondence: dhakal@sund.ku.dk (R.D.); hhh@sund.ku.dk (H.H.H.)

**Abstract:** Direct-fed microbial products (DFM) are probiotics that can be used advantageously in ruminant production. The in vitro gas production technique (IVGPT) is a method to simulate rumen fermentation and can be used to measure degradation, gas production, and products of fermentation of such additives. However, inter-laboratory differences have been reported. Therefore, tests using the same material were used to validate laboratory reproducibility. The objective of this study was to assess the effect of adding two DFM formulations on fermentation kinetics, methane ($CH_4$) production, and feed degradation in two different basal feeds while validating a newly established IVGPT laboratory. Six treatments, with three replicates each, were tested simultaneously at the established IVGPT lab at the University of Copenhagen, and the new IVGPT lab at Chr. Hansen Laboratories. Maize silage (MS) and grass silage (GS) were fermented with and without the following DFM: P1: *Ligilactobacillus animalis* and *Propionibacterium freudenreichii* (total $1.5 \times 10^7$ CFU/mL), P2: P1 with added *Bacillus subtilis* and *B. licheniformis* (total $5.9 \times 10^7$ CFU/mL). The DFM were anaerobically incubated in rumen fluid and buffer with freeze-dried silage samples for 48 h. Total gas production (TGP: mL at Standard Temperature and Pressure/gram of organic matter), pH, organic matter degradability (dOM), $CH_4$ concentration (MC) and yield (MY), and volatile fatty acid (VFA) production and profiles were measured after fermentation. No significant differences between the laboratories were detected for any response variables. The dOM of MS (78.3%) was significantly less than GS (81.4%), regardless of the DFM added (P1 and P2). There were no significant differences between the effects of the DFM within the feed type. MS produced significantly more gas than GS after 48 h, but GS with DFM produced significantly more gas at 3 and 9 h and a similar gas volume at 12 h. Both DFM increased TGP significantly in GS at 48 h. There was no difference in total VFA production. However, GS with and without probiotics produced significantly more propionic acid and less butyric acid than MS with and without probiotics. Adding P2 numerically reduced the total methane yield by 4–6% in both MS and GS. The fermentation duration of 48 h, used to determine maximum potential dOM, may give misleading results. This study showed that it is possible to standardize the methodology to achieve reproducibility of IVGPT results. Furthermore, the results suggest that the P2 DFM may have the potential to reduce $CH_4$ production without affecting organic matter degradation.

**Keywords:** methane ($CH_4$); in-vitro fermentation; direct-fed microbes; ruminants; probiotics; *Lactobacillus*; volatile fatty acid; fiber degradation; curve fitting

## 1. Introduction

Forage quality can limit production from ruminants through low voluntary feed intake and digestibility. The formulation of well-balanced rations to meet requirements is an important challenge in livestock production, as feeding can account for 60% of the cost [1]. Profitability in animal production depends, in part, upon the nutrient content

and digestibility of the feed. Feed digestibility is influenced by the physical and chemical properties of the feed.

Two techniques generally practiced for evaluating feed digestibility for ruminants are the in-vivo or in situ techniques (within the animal), and in-vitro techniques (outside the animal). In-vitro methods simulate the rumen fermentation conditions in a laboratory [2]. This allows for a significant amount of replication of an evaluation without the expense of many animals or large amounts of feed. At the same time, it allows for the collection of desired information ($CH_4$ emission, digestibility, volatile fatty acid (VFAs) production, and microbiota composition) at different time points with minimal influence on the experiment.

Rumen microbes anaerobically ferment fragmented feedstuffs and produce gases (mainly $CO_2$, $H_2$, and $CH_4$), VFAs, and microbial protein. Using the volume of gas produced during fermentation was proposed by Menke et al. [2,3] as a useful measure of the degradability and metabolizable energy content of feeds for ruminants. Therefore, the value of feed and additives can be estimated from the amount of gas produced during incubation in rumen fluid under anaerobic conditions [4]. Manual [2] or automatic methods [5,6] can be used to measure gas production. The products of fermentation can be collected during and after fermentation by collection in gas bags and filtration of the liquid and solid phases of the digesta.

Ring tests or validation tests, in which an experiment is repeated in another laboratory, are used to show the reproducibility of an experimental procedure. Studies and reviews [5,7] indicate the need for standardized procedures when comparing results from in-vitro gas fermentation systems. These studies have indicated differences in reproducibility between laboratories when using a similar methodology [5,8,9], but these studies did not use a standardized protocol, nor did they include measures of methane ($CH_4$) emission and VFA composition after fermentation.

Feed additives, containing non-pathogenic and non-toxic live microorganisms, have been used to improve animal performance and feed efficiency and to prevent disease in the livestock industry [10]. The use of direct-fed microbial products (DFM) involves feeding live microbes, beneficial to the animal, to improve health and performance [11]. The DFM consists of a group of microbes that can produce acids and bioactive compounds, compete against undesirable microbes, produce enzymes that stimulate desirable microbial growth, metabolize undesirable compounds, and/or stimulate the host animal's immune system [10–12]. Direct-fed microbes and microbial growth promoters have been studied for their ability to manipulate the microbial ecosystem and fermentation characteristics in the rumen and intestinal tract of livestock. Bacterial DFM has not only been shown to benefit the post-ruminal gastrointestinal tract but has also been found to play a beneficial role in the rumen itself by enzyme production and has potential benefits on feed fermentation and VFA production [12]. Because of this, there has been an increasing interest in using DFM in beef and dairy cattle diets [13]. Several *Bacillus sp.* have been shown to produce a wide set of fibrolytic, amylolytic, lipolytic, and proteolytic enzymes that can enhance the digestion in the rumen and increase the performance of the animals and are therefore used as probiotics [13]. Dias et al. [14] and Cull et al. [15] reported that the use of *Lactobacillus animalis* and *Propionibacterium freudenreichii* as DFM was effective in reducing the adverse effect of *Salmonella* infection in beef calves. Supplementation of *B. subtilis* improved growth performance and was beneficial to the intestinal microbiota in test animals [16]. Kan et al. [17] suggested *B. licheniformis* could be used as an antibiotic alternative against subclinical necrotizing enteritis.

Maize silage (MS) and grass silage (GS) are common feedstuffs in dairy cattle rations with distinctly different nutrient compositions. Maize silage contains more starch than grass silage, while grass silage contains more protein and fiber.

Two commercial DFMs contain *Bacillus* sp. have been evaluated for effects on nutrient utilization in dairy cows [18], but possible differential results of how these products act on grass and maize silage have not been clearly illustrated. It was of interest to see differences in fermentation kinetics, $CH_4$ production, and VFA production when using the chosen two

DFM formulations as feed additives to MS and GS while comparing results from the two laboratories using the same procedures and feedstuffs. Specifically, the first objective was to assess the intra-laboratory reproducibility of in-vitro gas production (IVGP) in relation to fermentation kinetics, total gas production, the concentration and yield of $CH_4$ and VFAs production and composition after 48 h of in-vitro fermentation. The second objective was to test differences between the effects of these DFM within and between the feeds on IVGP, fermentation kinetics, concentration and yield of $CH_4$, and VFA production, and composition in rumen fluid after 48 h of fermentation. We hypothesized that there would be no significant differences between laboratories when using a standard protocol and that both additives would increase organic matter degradation, while reducing $CH_4$.

## 2. Materials and Methods

### 2.1. Laboratories and In-Vitro System

Two laboratories were used for the research. The University of Copenhagen (KU) laboratory for feed evaluation has been using a semi-automatic in vitro gas production system (ANKOM[RF] Gas Production System, Macedon, NY, USA) since 2007. This same system was installed at the Chr. Hansen laboratory (Chr.) in 2018 and the KU protocol was used at both laboratories in this experiment. The ANKOM[RF] system releases pressure during a fermentation and constantly monitors gas pressure within multiple modules. The data sent from electronic chips embedded in the lid of a fermentation module is recorded on a computer and can be used to describe gas production kinetics. Gas tight bags (SKC, Flex Foil PLUS, Valley View Road, PN, USA) attached to the individual modules were used to collect the total gas produced during the incubation for analyses of gas composition.

### 2.2. Feeds, Treatments, and Rumen Fluid Donor Animals

The feeds used in the fermentation trials were maize silage (MS), collected in 2017, and grass silage (GS), collected in 2018. The forages were freeze-dried after harvest, ground in a cyclone mill (CT Cyclotex TM 193 TM, FOSS, Hillerød, Denmark) using a 2 mm sieve and stored until use. Final dry matter was determined before use by drying the freeze-dried samples in a forced-air oven (Memmert GmbH +Co. KG, Aeussere Rittersbacher Strasse, Schwabach, Germany) at 100 °C for 2 h and ash content was determined by burning the samples at 550 °C in a muffle oven (Carbolite Gero Ltd, Hope Valley, England) for 12 h and weighing the dried and burned samples after cooling to ambient temperature in a desiccator. Crude protein (CP) was determined by Kjeldahl nitrogen content using the VELP Kjeldahl system (VELP Scientifica, New York City, NY, USA). Fiber was determined after the principles of Van Soest [19]. Neutral detergent fiber with alpha amylase and without sulfite (aNDF) and acid detergent fiber (ADF) were determined using the protocol for the ANKOM Fiber Analyzer 200 (ANKOM, Rochester, NY, USA) [20]. Acid detergent lignin (ADL) was determined by the sulphuric acid method in a Daisy incubator (ANKOM, Rochester, NY, USA) according to the Daisy incubator lignin protocol [21]. Methodology details are described in Pandey et al. [22]. The chemical composition of the feed is presented in Table 1.

**Table 1.** Chemical composition of the feeds (% organic matter (OM) or % dry matter (DM)).

| Item | Maize Silage | Grass Silage |
|---|---|---|
| Dry matter % | 92.9 | 89.6 |
| Organic matter % | 95.7 | 91.3 |
| Crude protein% (DM) | 8.5 | 16.9 |
| aNDF % (DM) | 44.2 | 52.1 |
| ADF % (DM) | 24.1 | 28.3 |
| ADL % (DM) | 2.1 | 1.9 |
| Ash % (DM) | 4.3 | 8.7 |

aNDF: neutral detergent fiber with amylase, ADF: acid detergent fiber, ADL: acid detergent lignin, including acid insoluble ash, Ash: residue after burning in a muffle oven at 550 °C for 12 h.

The MS and GS with and without two DFM formulations (P1: *Ligilactobacillus animalis*, *Propionibacterium freudenreichii*, $1.5 \times 10^7$ total CFU/mL; P2: P1 + *Bacillus subtilis*, *Bacillus licheniformis*, $5.9 \times 10^7$ total CFU/mL) were used to test the hypotheses. A 0.5 g sample of one of the feeds (measured with 4 decimal accuracy) was weighed into three 100 mL bottles for each additive at each lab. This is a total of 18 bottles for each laboratory. All three treatments (control, P1, and P2) for each MS and GS were tested at each laboratory during two 48 h fermentations. Each fermentation at a given laboratory is considered a biological replicate, and the three bottles within each fermentation are considered technical replicates. One bottle with GS and no additive failed to register gas at the KU laboratory, leaving a total of only 11 bottles (a possible maximum of three bottles in each of two laboratories for two fermentations) for this treatment/feed combination. The doses used were those recommended by the manufacturer for commercial application. The treatments were named as follows: MS (control), MSP1 MSP2, GS (control), GSP1, GSP2. Additionally, three bottles with rumen fluid but no feed (BL) were included to determine the baseline fermentation.

Rumen fluid was collected at the University of Copenhagen Large Animal Hospital from two cannulated heifers that were fasted for 12 h. Fasting the animals is used in order to ensure that the microbial activity in the collected rumen fluid is stable with the lowest activity, and ass been reported earlier [22]. This helps to ensure that the fermentation products are a result of the feed and additives being tested. The use of cannulated animals was authorized according to Danish law (license nr.2012-15-2934- 00648). Animals were fed ad libitum haylage (85% dry matter, 7.5 MJ/kg metabolizable energy, and 11% crude protein per kg dry matter), for more than six weeks before the experiment. The rumen fluid collection procedure is described in detail by Vargas-Bello-Pérez et al. [23].

### 2.3. Experimental Procedures

A buffer solution was prepared as described by Menke and Steingass at each laboratory [2]. This medium, consisting of buffer, macrominerals, microminerals, and reazurin, was flushed with $CO_2$ at 39 °C for two hours before rumen fluid collection. Shortly before the arrival of the rumen fluid, a reduction agent of sodium sulfide and sodium hydroxide was added to ensure anaerobic conditions. The rumen fluid was added while flushing with $CO_2$, and the buffer media and rumen fluid temperature were maintained at 39 °C.

Two sets of rumen fluid with feed content were collected into warm thermos jars in equal amounts from each heifer. One was transported to the laboratory at KU and one to the Chr. laboratory for each fermentation. Transportation time to the KU and Chr. laboratories was approximately 35 min and 50 min, respectively. Upon arrival, the contents were strained through a double layer of commercial cheesecloth to remove coarse feed particles and gently squeezed to ensure microbial transfer from the particulate matter to the fluid. An equal amount of fluid from each heifer was used as inoculum for the in vitro incubation. The pH of the rumen fluid from each heifer and the pH of buffer media with rumen fluid were measured at the start of the experiment and from the residual liquid after filling the sample bottles.

Each sample bottle was filled with 90 mL of rumen fluid and flushed with $CO_2$ before closing the module and attaching a gas tight (SKC, Flex Foil PLUS, Valley View Road, Pennsylvania, U.S:A) sample bag to the vent valve tube of the module to collect all produced gas. A live recording interval of 60 s, a recording interval of 10 min, and a release pressure of 0.75 psi were used. The bottles were incubated in a ThermoShaker (Gerhardt, Königswinter, Germany) at 39 °C with 40 rotations per minute. At the end of the experiment, the sample bottle contents were filtered to collect undigested residue in a pre-weighed filter bag with a porosity of 25 μm (ANKOM F57, ANKOM Technology, Macedon NY, New York, country.S:A). The filter bags containing the undegraded residue were air dried at room temperature for 24 h, dried at 100 °C for 2 h, cooled to room temperature in a desiccator, and weighed.

The filtrate was collected from each bottle. A 0.8 mL aliquot was frozen at $-20\,°C$ for VFA analysis and pH measured in the remaining filtrate. Upon thawing, the VFA samples were re-filtered using a syringe filter with 0.2 μm porosity (MiniSart Syringe Filter, Satorius, Göttingen, Germany) and placed into a 7 mL test tube previously filled with 0.5 mL crotonic metaphosphoric solution. These solutions were mixed in a vortex for one minute and stored at $-20\,°C$ until analysis. VFA were analyzed using a gas chromatography withflame ionization detector (GC-FID)(Nexis GC-2030, Shimadzu Scientific Instruments Inc., Kyoto, Japan) at the University of Catania (Catania, Italy) as described by Carro et al. [24].

The $CH_4$ concentration in the gas-tight bags was measured directly after incubation by gas chromatography (GC) (Agilent 7820A GC, Agilent Technologies, Santa Clara, CA, USA). The GC was equipped with a HPPLOT Q column (30 m × 0.53 mm × 40 μmm), with $H_2$ as the carrier. The column flow was 5 mL/min and the thermal conductivity of the detector (TCD) was set to 250 °C with a reference and makeup flow of 10 mL/min. A 250 μL gas sample was taken from each gas bag and manually injected into the GC. The run time was 3 min at an isothermal oven temperature of 50 °C. Calibration curves were calculated from standards containing 1%, 2.5%, 5%, 10%, 15%, and 25% $CH_4$ in nitrogen (Mikrolab A/S, Aarhus, Denmark).

The data obtained from the BL bottles were used to correct gas production, $CH_4$ concentration, and VFA content for baseline values. After that, results from BL were not used, and only results of corrected variables are shown.

### 2.4. Calculations and Statistical Analyses

### 2.4.1. Calculations

The measured cumulative pressure (psi) was converted into gas volume (mL) by using the ideal gas law:

PV = nRT; where P = Pressure (PSI), V = Volume of gas (mL), n = moles of gas, R = gas constant, and T = Temperature (C) at standard temperature and pressure (STP)

The yield of gas (mL gas per gram of OM) was calculated from V, using the following formula:

$$\text{mL gas/g OM} = V/\text{g OM in the sample (corrected for baseline gas)}$$

Organic matter (OM) degradation was calculated as:

$$1 - ((\text{Final weight of the bag after fermentation} - (\text{empty bag weight, corrected for baseline OM microbial weight gains})/\text{OM in the sample})$$

$CH_4$ yield was calculated as follows:

$$CH_{4\,mL}\text{ per g OM} = (\text{Concentration of } CH_4 \text{ X Yield of gas})/100$$

### 2.4.2. Statistical Analysis

Statistical analyses were performed in R version 3.5.1 R Core Team (version: R4.0.05, Vienna, Austria) [25]. The 'drc' package [26] was used for curve fitting. Two sets of equations were used for the curve fitting of gas production (mL/min) as described by Dhakal et al., 2022 [27]: a sigmoidal curve [28] and three variations of a simple exponential curve [29]. The asymptote (A1) describes the maximum gas production, and the maximum slope (Vmax) and time at the maximum slope (Tmax) quantify the maximum rate of fermentation and when it occurs. The time at which half of the asymptote occurs (H1) quantifies when the fermentation is halfway to the maximum gas production. These parameters were extracted from the best fitting curve. The best fit was determined by the least Aikike information criteria (AIC).

The R function "lme" [30] was used to investigate differences in digestible organic matter and extract curve parameters in a linear mixed model. For all responses (OM, pH,

gas production at different time points, TGP, VFA, A1, H1, Tmax, Vmax and $CH_4$), the following model was used to compare the means of the treatments.

$$Yij = Treatment + Laboratory + (random= specific fermentation) + Error$$

where Yij is the value for the unit using treatment/feed combination (i) (GS, GSP1, GSP2, MS, MSP1, MSP2) from each laboratory (j). This model was stepwise reduced by removing statistically insignificant predictors. Differences among treatments were tested with Tukey's honestly significant differences. The normality of the residual error term was determined using the quantile-quantile plot of the residual using the R function qqnorm. Plots aligned with the theoretical normal line were assumed normal. Only the model for isovaleric acid had non-normal residuals and log transformed for normality. Significance was considered at $p < 0.05$ for all parameters except $CH_4$ production. A decreased sensitivity for significance was chosen for $CH_4$ ($p < 0.1$), because of previously determined variation in GC results.

## 3. Results

### 3.1. Effect of Laboratory

The best-fit curves were chosen by the least AIC for each treatment at each lab for the two fermentations, and the fitted curves for each sample and laboratory are shown in Figure 1. The sigmoidal model was the best fit for MS, MSP1, and MSP2. The exponential model with no intercept or lag time was the best fit for GS, GSP1, and GSP2. The shape of the gas production curves between the two laboratories was very similar. Maize silage curves, with or without additives, had a characteristically slower gas production than GS until between 11–13 h after the start of fermentation.

There were no significant differences ($p > 0.05$) between laboratories for any of the parameters measured: pH, dOM, total gas, concentration, or yield of $CH_4$ (Table 2), nor for H1, A1, Tmax, Vmax, and total VFA or VFA proportions (not shown). A large variation in the concentration of $CH_4$ was found in both laboratories, but no significant difference. Thereafter, the main effect of laboratory was removed from the model, but "laboratory" was included as a random effect.

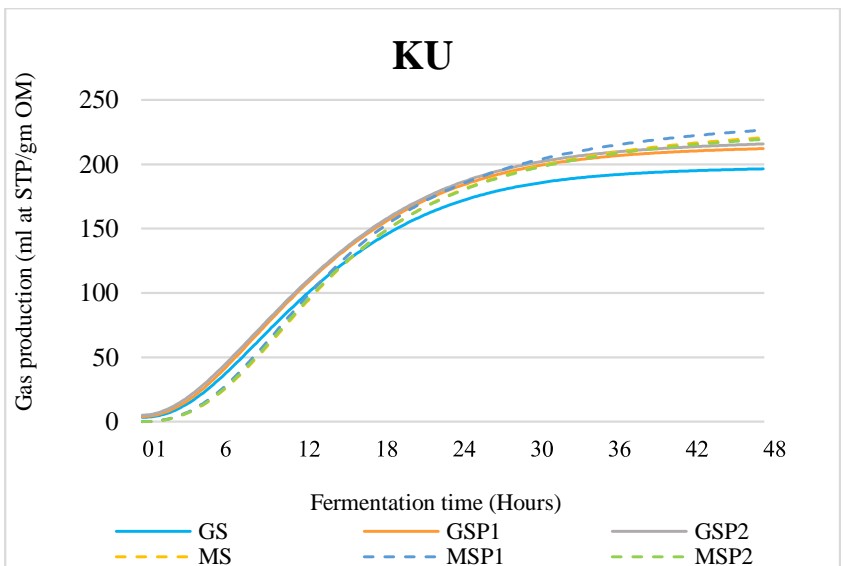

**Figure 1.** *Cont*.

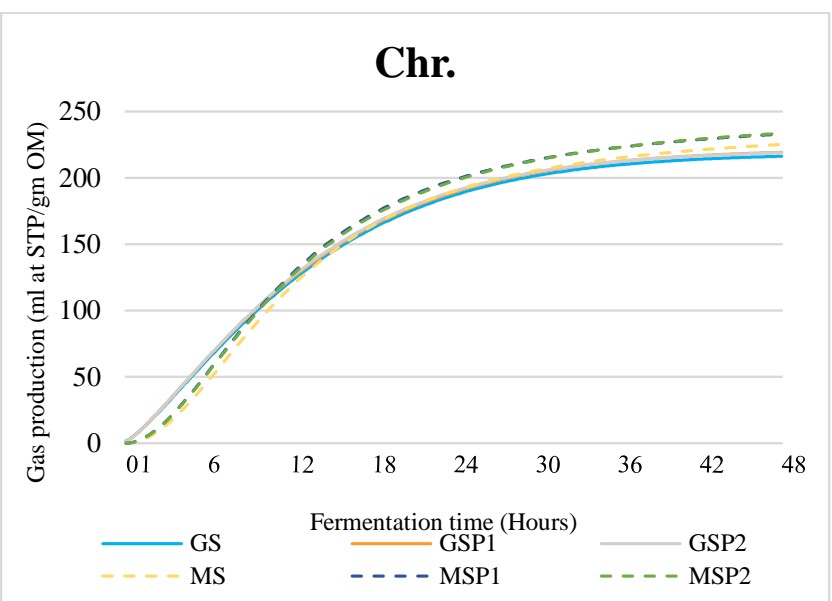

**Figure 1.** Cumulative gas production of fitted curves from data from fermenting grass silage (GS) and maize silage (MS) with or without probiotic 1 (P1) or probiotic 2 (P2) in rumen fluid for 48 h at two laboratories. Chr.: Chr. Hansen, KU: University of Copenhagen, STP: Standard temperature and pressure, OM: Organic matter.

**Table 2.** Laboratory results for pH, organic matter degradation (dOM), total gas production (TGP), $CH_4$ yield, and $CH_4$ concentration after 48 h of in-vitro fermentation of grass silage (GS) and maize silage (MS) with or without probiotic 1 (P1) or probiotic 2 (P2).

| | GS | | GSP1 | | GSP2 | | MS | | MSP1 | | MSP2 | | SEM |
|---|---|---|---|---|---|---|---|---|---|---|---|---|---|
| | Chr. | KU | Chr. | KU | Chr. | KU | Chr. | KU | Chr. | KU | Chr. | KU | |
| pH | 6.8 | 7.0 | 6.9 | 7.0 | 6.9 | 7.0 | 6.8 | 6.9 | 6.8 | 6.9 | 6.8 | 7.0 | 0.047 |
| dOM | 88.8 | 89.5 | 88.7 | 88.8 | 91.9 | 88.8 | 81.4 | 82.3 | 81.4 | 81.8 | 80.7 | 82.1 | 0.72 |
| TGP (48 h) | 217.6 | 197.5 | 219.6 | 212.6 | 219.0 | 215.6 | 224.6 | 220.6 | 231.8 | 226.3 | 232.8 | 219.0 | 6.46 |
| $CH_4$ concentration (%) | 7.2 | 9.9 | 8.8 | 8.8 | 7.7 | 8.0 | 10.6 | 9.9 | 10.7 | 9.1 | 11.3 | 7.7 | 1.78 |
| $CH_4$ Yield (mL/g OM) at STP | 15.5 | 19.5 | 19.2 | 18.7 | 16.7 | 17.3 | 23.8 | 22.0 | 25.0 | 20.5 | 26.2 | 17.0 | 0.73 |

SEM: standard error of the mean, MS: maize silage, GS: grass silage, GSP1: grass silage with probiotic 1, GSP2: grass silage with probiotic 2, MSP1: maize silage with probiotic 1, MSP2: maize silage with probiotic 2, Chr.: Chr. Hansen, KU: University of Copenhagen.

### 3.2. Effect of Probiotics

3.2.1. Feed Organic Matter Degradation

The organic matter degradation of the different treatments is presented in Table 3. There was a significant difference ($p < 0.05$) in the degradation of MS versus GS, regardless of the additives (P1 and P2). However, there were no significant ($p > 0.05$) effects of the DFM within feed type. The pH values of the rumen fluid obtained from the heifers were 7.26 and 6.98 upon arrival at KU, and 7.16 and 7.05 upon arrival at Chr. for the first and second experiments, respectively. The pH value of the fluids of MS and GSP2 was significantly different after fermentation.

**Table 3.** Organic matter degradation (%) and pH after 48 h of fermentation of grass silage (GS) and maize silage (MS) with or without probiotic 1 (P1) or probiotic 2 (P2) in rumen fluid (data from 2 laboratories).

| Treatment | MS | MSP1 | MSP2 | GS | GSP1 | GSP2 | SEM |
|---|---|---|---|---|---|---|---|
| N | 12 | 12 | 12 | 11 | 12 | 12 | |
| Organic matter degradation (%) | 78.29 [a] | 78.02 [a] | 77.87 [a] | 81.42 [b] | 81.09 [b] | 82.51 [b] | 0.72 |
| pH | 6.87 [a] | 6.88 [ab] | 6.88 [ab] | 6.91 [ab] | 6.91 [ab] | 6.92 [b] | 0.047 |

[a,b] Values within a row are different if superscript differs ($p < 0.05$), N: number of observations, SEM: standard error of the mean, MS: maize silage, MSP1: maize silage with probiotic 1, MSP2: maize silage with probiotic 2, GS: grass silage, GSP1: grass silage with probiotic 1, GSP2: Grass silage with probiotic 2.

### 3.2.2. Total Gas Production

The total gas production at chosen time points is shown in Table 4. MS and GS feeds without probiotics produced significantly different total gas at 3, 9, and 48 h. MS produced significantly less gas at 3 and 9 h but GS produced less at 48 h. No significant differences between MS and MS with additives or between the additives in MS were seen at any time. However, GSP2 produced significantly more gas than GS at 3 h and both GSP1 and GSP2 produced more gas than GS at 48 h but were not different from each other.

**Table 4.** Total gas production (mL gas at STP per gram organic matter) at a chosen time points during fermentation of grass or maize silage with or without probiotic 1 (P1) or probiotic 2 (P2) (pooled data from 2 laboratories).

| Time/Treatment | MS | MSP1 | MSP2 | GS | GSP1 | GSP2 | SEM |
|---|---|---|---|---|---|---|---|
| N | 12 | 12 | 12 | 11 | 12 | 12 | |
| 3 h | 18.67 [a] | 20.56 [ab] | 19.85 [ab] | 22.55 [bc] | 25.52 [cd] | 27.89 [d] | 4.68 |
| 9 h | 60.3 [a] | 66.61 [a] | 64.89 [a] | 74.95 [b] | 80.24 [b] | 81.09 [b] | 11.17 |
| 12 h | 102.62 | 109.71 | 107.65 | 105.13 | 110.85 | 110.91 | 9.05 |
| 48 h | 222.59 [bd] | 229.03 [d] | 225.93 [cd] | 205.68 [a] | 216.10 [b] | 217.32 [bc] | 6.46 |

[a,b,c,d] Values within a row are different if superscripts differ ($p < 0.05$), SEM: standard error of the mean, MS: maize silage, MSP1: maize silage with probiotic 1, MSP2: maize silage with probiotic 2, GS: grass silage, GSP1: grass silage with probiotic 1, GSP2: grass silage with probiotic 2, N: number of observations.

### 3.2.3. Fitted Curve Parameters of Total Gas Production

The extracted parameters from the fitted gas curves are shown in Table 5. Total maximum gas production for GS, GSP1, or GSP2 was significantly less and took less time to reach half of the maximum than MS (MS, MSP1, and MSP2) for all parameters. The maximum gas production occurred before the end of fermentation for both feeds and additive combinations. Regardless of treatments, GS produced half of the theoretical maximum accumulated gas production before (11 h) MS with or without probiotics, which took over 13 h, but the significantly faster Vmax of MS, MSP1 and MSP2 resulted in significantly more gas at the mathematical asymptote and end of fermentation at 48 h. There were no significant differences between the MS treatments for A1, H1, Vmax, and Tmax. Both DFM treatments in GS had a significantly greater theoretical maximum gas production (A1) than GS.

**Table 5.** Fitted curve parameters (pooled data from 2 laboratories).

| Feed | MS | MSP1 | MSP2 | GS | GSP1 | GSP2 | SEM |
|---|---|---|---|---|---|---|---|
| N | 12 | 12 | 12 | 11 | 12 | 12 | |
| A1 (mL gas STP/g OM) | 237.75 [c] | 245.17 [c] | 241.96 [c] | 207.81 [a] | 219.57 [b] | 222.46 [b] | 6.78 |
| H1 (h) | 13.83 [b] | 13.51 [b] | 13.63 [b] | 10.97 [a] | 10.97 [a] | 10.99 [a] | 0.98 |
| Vmax (mL gas STP/g OM) | 2.03 [b] | 2.11 [b] | 2.08 [b] | 1.81 [a] | 1.89 [a] | 1.89 [a] | 0.15 |
| Tmax (h) | 8.92 [b] | 8.56 [b] | 8.63 [b] | 7.25 [a] | 6.83 [a] | 6.87 [a] | 1.64 |

[a,b,c] Values within a row are different if superscripts differ ($p < 0.05$); N: number of observations. A1: asymptotic mL/g OM, H1: time at which half of the asymptotic amount is accumulated, Vmax: maximum slope and Tmax: time of maximum slope, MS: maize silage, MSP1: maize silage with probiotic 1, MSP2: maize silage with probiotic 2, GS: grass silage, GSP1: grass silage with probiotic 1, GSP2: grass silage with probiotic 2, STP: Standard temperature and pressure, OM: Organic matter.

3.2.4. Methane ($CH_4$) Concentration and Yield

The concentration and yield of $CH_4$ produced when fermenting MS and MSP1 was significantly ($p < 0.1$) greater than GSP2 which produced the absolute least concentration and yield of $CH_4$ (Table 6). However, there were no differences between GS and GS treated with probiotics nor the MS and MS treated with probiotics.

**Table 6.** Total $CH_4$ yield (ml STP/g OM) and concentration after 48 h in-vitro rumen fermentation of grass and maize silage with or without probiotic 1 (P1) or probiotic 2 (P2) (data from 2 laboratories).

| | MS | MSP1 | MSP2 | GS | GSP1 | GSP2 | SEM |
|---|---|---|---|---|---|---|---|
| N | 12 | 12 | 12 | 11 | 12 | 12 | |
| $CH_4$ Yield (mL STP/g OM) | 22.90 [b] | 22.74 [b] | 21.58 [ab] | 17.67 [ab] | 18.94 [ab] | 16.97 [a] | 1.78 |
| $CH_4$ Concentration (%) | 10.28 [b] | 9.90 [ab] | 9.49 [ab] | 8.63 [ab] | 8.77 [ab] | 7.84 [a] | 0.73 |

[a,b] Values within a row are different if superscripts differ ($p < 0.1$), SEM: standard error of the mean, MS: maize silage, MSP1: maize silage with probiotic 1, MSP2: maize silage with probiotic 2, GS: grass silage, GSP1: grass silage with probiotic 1, GSP2: grass silage with probiotic 2, STP: standard temperature and pressure, N: number of observations.

There was no significant difference ($p > 0.05$) in total VFA production between any of the treatments (Table 7), but grass silage with probiotics P2 (GSP2) produced numerically the most VFA (mmol/L) and MS produced the least of all treatments.

**Table 7.** VFA production and composition after 48 h of in-vitro rumen fermentation of grass and maize silage with or without probiotic 1 (P1) or probiotic 2 (P2) (data from 2 laboratories).

| | MS | MSP1 | MSP2 | GS | GSP1 | GSP2 | SEM |
|---|---|---|---|---|---|---|---|
| Total (mmol/L) | 33.07 | 34.38 | 32.92 | 33.39 | 32.92 | 34.71 | 1.41 |
| Acetic (% of Total) | 54.31 [abc] | 53.27 [a] | 53.63 [ab] | 59.84 [cd] | 60.55 [d] | 59.0 [cd] | 2.44 |
| Propionic (% of Total) | 21.95 [a] | 22.29 [a] | 22.05 [a] | 25.65 [b] | 25.63 [b] | 26.30 [b] | 0.79 |
| Isobutyric (% of Total) | 0.97 | 1.02 | 0.98 | 1.02 | 1.02 | 0.95 | 0.13 |
| Butyric (% of Total) | 18.94 [b] | 19.28 [b] | 19.48 [b] | 9.26 [a] | 8.87 [a] | 9.58 [a] | 1.23 |
| Isovaleric (% of Total) | 1.77 | 1.85 | 1.80 | 1.50 | 1.40 | 1.54 | 0.12 |
| Valeric (% of Total) | 1.41 [a] | 1.48 [a] | 1.39 [a] | 2.14 [b] | 2.06 [b] | 2.18 [b] | 0.18 |
| Caproic (% of Total) | 0.54 | 0.67 | 0.58 | 0.39 | 0.40 | 0.38 | 0.23 |
| Acetic: Propionic | 2.47 | 2.39 | 2.43 | 2.33 | 2.36 | 2.24 | 0.16 |

[a,b,c,d] Values within a row are different if the superscript differs ($p < 0.05$), SEM: standard error of the mean, MS: maize silage, MSP1: maize silage with probiotic 1, MSP2: maize silage with probiotic 2, GS: grass silage, GSP1: grass silage with probiotic 1, GSP2: grass silage with probiotic 2.

Volatile fatty acid (VFA) production differences were found between feeds but not between feeds with and without DFM treatments. Acetic acid and propionic acid production were greater in GS, with and without DFM, compared to MS with or without the same feed additives. There were no significant differences between the treatment/feed

combinations for the production of isobutyric, isovaleric, and caproic acids, or the ratio of acetic acid to propionic acid. Butyric acid production from GS with and without DFM (GSP1 or GSP2) was significantly less than from MS with and without DFM, whereas valeric acid production from GS and GS with DFM (GSP1 or GSP2) was significantly greater than MS and MSP1 and MSP2.

## 4. Discussion

### 4.1. Effect of Laboratory

The in vitro gas production (IVGP) technique has been used widely to measure total gas, $CH_4$ concentration, and yield [2,31]. However, the limitation of the IVGP methodology used in this research was the necessity of balancing the inability to address passage rate and the choice of endpoint for feed degradation. There is no passage rate in a closed bottle batch system, and therefore results can be dependent on the choice of endpoint fermentation time. The choice of 48 h for fermentation should allow sufficient time for potential maximum fiber digestion but may be unrealistic in high-yielding dairy cows. According to the NORFOR feed system [32], the retention times for Danish dairy cows are around 6, 16, and 37 h, respectively, for the liquid phase, the protein and starch fraction, and the NDF fraction. A 48 h fermentation was desired to obtain sufficient data regarding the effect of the DFM on dOM.

### 4.2. Effect of DFM

Although significant, the differences between GS and MS dOM with or without probiotics of between 4 and 6 % and may not be of a magnitude that will occur in vivo with a more rapid passage rate. The time to reach half of the maximum gas was significantly faster for GS with and without additives than MS but over 80% of the total gas was produced in the first 24 h for both MS and GS. As with the dOM differences at 48 h, the gas increases between 24 and 48 h may only reflect a theoretical potential for fiber retention and further degradation but may not reflect a passage rate in high-yielding dairy cows. Finally, the significant increase in valeric acid in the GS samples suggests microbial lysis [33]. This too, suggests that the results at 48 h may not reflect high-yielding dairy cow digestion.

The findings from this study showed that it was possible to achieve similar results for IVGP in two laboratories using the same protocol, feed, and rumen fluid. All in-vitro fermentation parameters, and VFA total production, and composition concentrations in the rumen fluid after fermentation were similar between laboratories. This suggests that the IVGP method can be standardized to achieve reproducible results.

There were no pH differences between MS and GS at 48 h of fermentation, and the pH of MS was only significantly less than GSP2. Acetic acid is a stronger acid than butyric acid which, in turn, is stronger than propionic acid. While the proportions of acetic and propionate were significantly greater by between 11 and 18% in GS, GSP1, and GSP2 compared to MS, MSP1, and MSP2, the butyric acid concentration was between 2.03 and 2.17 times greater in MS, MSP1, and MSP2 than in GS, GSP1, and GSP2. The molar ratio of propionate and acetate was 1.2 and 1.1 times greater in GS as compared to MS. This should have resulted in a lower pH in the GS but was not observed. However, the lower pH in the GS would be mitigated by the over two times greater molar ratio of butyric acid in MS compared to GS rendering the difference non-significant.

Adding DFM (P1 and P2) to both MS and GS increased total gas production compared to pure MS and GS. This difference was significant at 48 h for GS but not for MS. This agrees with previous research that has shown that the addition of DFM, and *Ligilactobacillus animalis* (formerly *Lactobacillus* sp.) in particular, increases gas production [34,35], and this could be due to an increase in microbial growth in the feeds with additives. The fact that a difference was not seen before 48 h could be that the added proteins are complex macromolecules, and they are more difficult to break down and utilize than other simple carbon sources like sugars for many of the rumen microorganisms. The rumen microbes might need a long time to adapt to the additives in order to produce the enzymes required

to break down the protein source. However, this may not be a realistic time frame for the high-yielding dairy cow.

The dietary addition of probiotic microorganisms can modulate the balance and activities of the gastrointestinal microbiota and strongly affect the structure and activities of the gut microbial communities [11]. Studies have also shown a beneficial impact of probiotics on animal productivity by affecting the stabilization of the rumen environment and increasing fiber degradation during fermentation [14,36,37]. This results in increased nutrient availability and utilization, increased milk production, and animal growth [38–41]. The lack of significant difference found in the present research does not reflect these earlier studies. This was unexpected, but not indicative that the differences exist, and suggests the need for more further validation of the results.

The chemical composition of the feed is an important factor for predicting the degradability of dry matter or organic matter in in-vitro gas production [42]. In theory, the greater the OM degradability, the greater the gas production will be. The findings from our study agree with this theory at 3 and 9 h. However, the findings from of our study showed a greater dOM yet less gas production at 48 h from GS, GSP1, and GSP2 compared to MS, MSP1, and MSP2. There was an average of 24 % more gas in the GS samples at 3, and 9 h, no difference at 12 h but only an 11% difference in the GS samples compared to the MS samples at 48 h. This shows that the GS gas production stagnated while the MS continued to produce gas.

During fermentation, the rumen microbiome plays a vital role in total gas production, $CH_4$ production, and VFA concentration. Methane ($CH_4$) production is known to be influenced by the addition of probiotics, additives, or different supplements in the feed [43–45]. We found no significant differences in $CH_4$ yield in both silages with the additives compared to the pure silage, and no significant differences between the silages without additives. The $CH_4$ results indicated that there was no effect of the additives, but this is not in agreement with other research [13,35–37,39]. The P2 is a mix of *Ligilactobacillus animalis*, *Propionibacterium freudenreichii* with additional *Bacillus subtilis* and *Bacillus licheniformis* and has a greater concentration of CFU's, leading to an expectation of more microbial activity. An increased microbial activity would occur if the additional microbes in P2 increase the load of CFU's beyond a threshold value which is needed to reduce $CH_4$.This was seen in previous research [35–37,46]. Alternatively, the lack of difference may be caused by a $CH_4$ increasing effect of *Ligilactobacillus animalis* and *Propionibacterium freudenreichii*.

The microbes in P2 have been shown to increase fiber degradation in the rumen through increasing microbial enzyme production, thereby modulating VFA production in-vitro and in the rumen [12,13]. A change in fiber degradation will lead to a change in VFA production and this can directly modify the rumen microbial populations and $CH_4$ production. However, the VFA profile and production did not differ significantly in this research. The use of DFM as probiotics has previously been seen to improve in-vitro fermentation characteristics [35–37] and increase ruminal propionate concentration [47]. The addition of *Lactiplantibacillus plantarum (L. plantarum)* with cellulase enzyme in *Caragana korshinskii* silage was seen to increased NH3-N and microbial protein content and increased degradation rate of dry matter during in vitro fermentation [48]. According to Maderal et al. [49], the increasing dose of *Bacillus sp.* decreased the molar proportion of acetate and increased the molar proportion of propionate, and the rate of gas production quadratically increased. Despite these differences, the total gas production and $CH_4$ emission was not affected in this research. This lack of TGP and $CH_4$ production differences agree with the results from the present study.

Total gas production is affected by the type of feed, microbial protein synthesis, the chemical composition of substrate used and the proportions of VFA produced [50,51] during the fermentation of feeds. During fermentation, acetate and butyrate are the main source of gas production [31,52–54]. Feeds with greater contents of structural carbohydrates digest slower and produce less gas because of decreased microbial activity. Degradation of easily digestible carbohydrates, as opposed to structural carbohydrates, produces more

total VFA with a greater proportion of propionate. Relatively less hydrogen gas is produced when propionate production increases. The production of propionate is a hydrogen sink, in that it utilizes two molecules of hydrogen whereas acetate yields 2 molecules of carbon dioxide and 4 molecules of hydrogen, and the production of butyrate yields 2 molecules of carbon dioxide and two molecules of hydrogen [53,55]. The proportion of propionic acid in the rumen fluid after fermentation with or without P1 or P2 was significantly greater in GS than in all MS treatments, in direct contrast to the greater content of structural carbohydrates in the grass silage. However, this was observed even though the ratio of acetic to propionic acid was not significantly different between treatments. The increase in propionate, but constant acetate to propionate ratio among treatments indicates no real change in the hydrogen balance and therefore no overall change in the total gas production due to changes in hydrogen consumption or production.

Dietary alteration can influence the entire rumen microbiome affecting the methanogens within the rumen ecosystem [56]. Changes in diet can be caused by different feed sources, additives, or DFM. The tested DFM, in the given dosages, showed no significant differences in enteric $CH_4$ production, and therefore further research is needed to understand the reasons for the mitigating effects found elsewhere but not in our research. The use of a donor cow that has been adapted to the probiotics should be investigated, as well as the synergistic effects of the additives and different application doses.

## 5. Conclusions

This study showed that it was possible to use a standardized protocol and achieve reproducible results for in-vitro gas production research in two separate laboratories. The total gas production, curve parameters, organic matter degradability, $CH_4$ concentration and yield, and total VFA production and composition in the rumen fluid were similar between the two laboratories. The expected differences when adding the DFM were not supported by the research, but the duration of fermentation may have been a detrimental factor. The two tested direct fed microbials showed non-significant results that could indicate greater degradability in GS at shorter fermentation times, and the combination of *L. animalis*, *P. freudenreichii*, *B. licheniformis*, and *B. subtilis* and showed a non-significant reduction of $CH_4$ yield when added to both types of feeds. To better understand the mode of action of the tested additives on rumen feed degradation, differing application doses and incubation durations are suggested.

**Author Contributions:** Conceptualization R.D., G.C., B.I.C. and H.H.H.; methodology and investigation, R.D., N.M., G.C., B.I.C. and H.H.H.; formal analysis, data curation, writing-original draft preparation, R.D.; writing-review and editing, R.D., G.C., B.I.C. and H.H.H.; supervision, H.H.H.; funding acquisition, H.H.H. and G.C. All authors have read and agreed to the published version of the manuscript.

**Funding:** This research was funded by Chr. Hansen, Animal and Plant Health & Nutrition. Bøge Alle 10-12, 2970 Hørsholm, Denmark.

**Institutional Review Board Statement:** The use of cannulated animals was authorized according to Danish law (license nr.2012-15-2934- 00648).

**Informed Consent Statement:** Not applicable.

**Data Availability Statement:** The data will be available upon request to the corresponding authors.

**Acknowledgments:** Many thanks to the assistance from laboratory technicians Anni Christiansen and Lotte Ørbæk.

**Conflicts of Interest:** The authors declare no conflict of interest.

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
