# Peer review of "The Effect of Direct-Fed Microbials on In-Vitro Rumen Fermentation of Grass or Maize Silage"

_fermentation, doi:10.3390/fermentation9040347_

Round 1
Reviewer 1 Report
The review report is attached.

Author Response
Thank you very much for reviewing our manuscript. We have tried to address all the comments and suggestions.

Reviewer 2 Report
This manuscript described the influence of two direct-fed microbials on in vitro rumen fermentation characteristics of grass and maize silages, which is not new, but falls into the scope of journal Fermentation. However, the authors should deal with the major comments below before further consideration for acceptance.
First, the authos did not normatively use the units in the text. For instance, using "hours" rather than "h" (L28-29, 33, 104, 107, 129-129, 148, 158, 191, 259, 272, 286, 294-298, 312, 314, 329,359, 362, 365-367, 371-373, 376, 382, 390, 392, 412-413 and 415); using "minutes" rather than "min" (176, 186, 188 and 197); using "second" rather than "s" (185-186); "Bacillus spp." should be "Bacillus sp."; Pandey et al. (L137), Dhakal et al. (L229), "et al." should not be italic.
Second, the authors should keep in accordance between the full words and their abbreviations throughout the tex, for instance, methane and CH4, VFA and volatile fatty acid.
Third, in "Discussion" section, it shoud keep in accordance with the structure of 'Results" section, namely the "Discussion" section should consist of 4.1 Effect of laboratoray and 4.2 Effect of probiotics (4.2.1, 4.2.2, 4.2.3 and 4.2.4).
Last, the Latin names of creatures in the Reference shouod be italic (L530, 532, 537, 540, 542, 578, 584, 591, 593 and 596); the journal name should be abbreviated and the volume number should be italic.
Author Response

(The authors gave the same response as above.)

Round 2
Reviewer 1 Report
Most of the questions were answered and errors were corrected. I had just a few comments.

Reviewer 2 Report
None